# Amplifying the Output of a Triboelectric Nanogenerator Using an Intermediary Layer of Gallium-Based Liquid Metal Particles

**DOI:** 10.3390/nano13071290

**Published:** 2023-04-06

**Authors:** Jong Hyeok Kim, Ju-Hyung Kim, Soonmin Seo

**Affiliations:** 1Department of Bionano Technology, Gachon University, Seongnam 13120, Republic of Korea; whdgur64@gmail.com; 2Department of Chemical Engineering and Department of Energy Systems Research, Ajou University, Suwon 16499, Republic of Korea

**Keywords:** triboelectric nanogenerators, TENG, liquid metal, Galinstan, liquid metal particles, Galinstan particles, dielectrics

## Abstract

The production of energy has become a major issue in today’s world. Triboelectric nanogenerators (TENGs) are promising devices that can harvest mechanical energy and convert it into electrical energy. This study explored the use of Galinstan particles in the production of TENGs, which convert mechanical energy into electrical energy. During the curing process, the evaporation of the hexane solvent resulted in a film with varying concentrations of Galinstan particles. The addition of n-hexane during ultrasonication reduced the viscosity of the polydimethylsiloxane (PDMS) solution, allowing for the liquid metal (LM) particles to be physically pulverized into smaller pieces. The particle size distribution of the film with a Galinstan concentration of 23.08 wt.% was measured to be within a few micrometers through ultrasonic crushing. As the amount of LM particles in the PDMS film increased, the capacitance of the film also increased, with the LM/PDMS film with a 23.08% weight percentage exhibiting the highest capacitance value. TENGs were created using LM/PDMS films with different weight percentages and tested for open-circuit voltage, short-circuit current, and charge amount Q. The TENG with an LM/PDMS film with a 23.08% weight percentage had the highest relative permittivity, resulting in the greatest voltage, current, and charge amount. The use of Galinstan particles in PDMS films has potential applications in wearable devices, sensors, and biomedical fields.

## 1. Introduction

With the widespread use of mobile devices in daily life, there is growing interest in developing smaller, more functional, and portable devices. Various methods of power generation have been adopted to operate these mobile devices. However, environmental pollution caused by energy generators is a concern, and researchers are investigating sustainable energy generation methods. One such technology is energy harvesting, which involves collecting energy from natural phenomena such as solar power, wave power, and wind power and converting it into electrical energy. Triboelectricity, which is the electricity generated when different substances rub against each other, has gained considerable attention in recent years [1,2,3,4,5]. Due to the different dielectric constants of each surface material that causes friction, the amount of energy generated also varies. Many studies are being conducted to improve the performance of triboelectric generators by developing new materials or using materials with dielectric constants that have different levels of polarity to produce friction in the generator. Researchers are also attempting to add impurities to dielectric films, change surface properties, or adjust the thickness of thin films to generate high amounts of energy [6,7,8,9,10,11].

Research on frictional power generation devices has explored the potential applications of wearable, flexible, and patchable devices [12]. Liquid metals (LMs) have emerged as a promising material for creating flexible devices due to their low melting points and conductivity, making them suitable for use as electrodes in stretchable devices that utilize polymers [13,14]. While mercury is a commonly known LM, its toxic properties have prompted researchers to explore gallium-based metal alloys with lower toxicity. Galinstan, a gallium-based metal alloy composed of 68 wt.% gallium, 22 wt.% indium, and 10 wt.% tin, has been of particular interest due to its low toxicity and ability to exist in a liquid state even at temperatures below the freezing point of water [15]. This gallium-based LM can be processed in a vacuum chamber and does not require a hood, making it advantageous for creating flexible devices.

Recent studies have investigated various flexible devices associated with polydimethylsiloxane (PDMS) [16,17,18], but there have been no reported cases of using a thin film of LM in PDMS for triboelectric nanogenerators (TENGs). Although inserting LM particles into the dielectric material may improve the TENG performance, forming a thin film with an LM has proven difficult due to the high surface energy of the LM. The use of dielectrics with LM particles has the advantage of reversibility, as the dielectric can return to its initial state even after deformation caused by external stimuli is removed, compared to dielectrics with solid fillers inserted. Additionally, another advantage of using LM particles as fillers in dielectrics is the simplicity of the production process, which can reduce the processing time and production costs. The liquid metal in bulk form can be mixed with PDMS and n-hexane solvents and crushed into particles through ultrasonication, making them easy to insert into the dielectric film. Furthermore, creating and distributing LM particles within the high-viscosity PDMS material have posed challenges.

To address these challenges, this study proposes a method of producing LM particles through ultrasonication while adopting a conductor–dielectric mode of a TENG with a vertical contact-separation structure. This study fabricated a TENG device by inserting layers of Galinstan particles with varying concentrations into a PDMS film and improved its electrical characteristics through friction with copper. The study compared and analyzed voltage, current, and capacitance measurements to investigate the impact of LM particles on the TENG performance when inserted into a PDMS dielectric material.

## 2. Materials and Methods

### 2.1. Fabrication of a TENG Using PDMS Embedded with Galinstan Particles

A reference sample was prepared using pristine PDMS without any additives. For the test samples, base PDMS and a curing agent were mixed at a ratio of 10:1 wt.% and spin-coated at 500 rpm for 30 s to prepare a thin film (190 μm). For the PDMS film embedded with Galinstan particles, 10 g of n-hexane was added to a conical tube, and 2 g of PDMS and 0.2 g of curing agent were added. Then, 0.05 g, 0.12 g, 0.32 g, 0.66 g, and 1.49 g of Galinstan were added to each mixture, and they were ultrasonicated for 10 min at 200 W using a sonicator (KUS-650, KBT, Seongnam, Republic of Korea). The resulting weight percentages of Galinstan in the prepared samples were 0.41 wt.%, 0.97 wt.%, 2.55 wt.%, 5.13 wt.%, and 10.88 wt.%, respectively. The mixture was poured into Petri dishes with a diameter of 150 mm and evaporated with hexane for 24 h in a 50 °C oven. The Galinstan particles quickly precipitated and formed a sediment layer upon pouring. In addition, the PDMS was then further cured by leaving it in a 70 °C oven for 2 h. After cutting the cured PDMS film into an area of 2.5 cm × 2.5 cm, the PDMS film was flipped over and spin-coated with a 10:1 ratio of PDMS at 1500 rpm for 30 s. The uncured PDMS mixed with the cured PDMS film combined after curing and the Galinstan particles were positioned in the middle layer of the film, as shown in Figure 1b. In this way, samples with an area of 2.5 cm × 2.5 cm and a total thickness of 190 μm were prepared. The prepared PDMS film was attached to a slide glass with copper tape and fixed with conductive carbon tape.

To confirm the evaporation of hexane, pristine PDMS was prepared using the same conditions, and hexane (10 g) was mixed with PDMS (10:1) at a quantity of 2.2 g. This mixture was subjected to hexane evaporation at 50 °C in an oven for 24 h. The partially cured PDMS was then further cured by leaving it in a 70 °C oven for 2 h. Each film was subsequently cut into an area of 2 cm × 2 cm. The volatile nature of hexane, used in the production of LM particles, and the permeability of PDMS result in the evaporation of hexane during the curing process. As a result, the n-hexane solvent is almost entirely absent from the LM/PDMS film after PDMS curing. In this study, the weight ratios of the LM and PDMS were 0.41 wt.%, 0.97 wt.%, 2.55 wt.%, 5.13 wt.%, and 10.88 wt.%. However, after the evaporation of hexane, the remaining amounts of PDMS and Galinstan were used to recalculate the weight ratio. The weight percentages of Galinstan in these samples were 2.22 wt.%, 5.17 wt.%, 12.97 wt.%, 23.08 wt.%, and 40.30 wt.%, based solely on the weight of the PDMS, curing agent, and Galinstan. 

### 2.2. Analysis of Electrical Characteristics of TENGs

#### 2.2.1. Measurement of Voltage and Current According to Weight Percentage

To investigate the amplification of triboelectricity generation by the Galinstan particle layer, the voltage and current were measured according to the weight percentage. For the conductor–dielectric configuration, the electrode was created by attaching copper tape to the relative component. For the vertical contact-separation configuration, an instrument using a linear motor was manufactured to ensure that the conditions were fixed, enabling the component to be contacted at a constant speed and pressure. Figure 1c depicts the device setup used to measure the performance of the TENG. The top electrode was connected to a linear motor, while the liquid metal-PDMS (LM/PDMS) on copper electrode was connected to the opposite force gauge stage, maintaining a constant force of 1N while undergoing oscillatory motion at a frequency of 5 Hz and a distance of 55 mm. The voltage was measured with an oscilloscope (DS1104, RIGOL, Portland, OR, USA), and the current was measured with an electrometer (Keithley-6514, Keithley, Cleveland, OH, USA) for each sample. Since it is not possible to measure the size of LM particles using SEM, they were measured using an optical microscope (Eclipse LV-150N, NIKON, Tokyo, Japan). The black areas of the LM particles observed under the microscope with a backlight were measured using ImageJ software, and the radius was calculated using the measured values.

#### 2.2.2. Capacitance Measurement

In order to investigate the cause of amplification resulting from the presence of the intermediate layer of Galinstan, the capacitance was measured using thin LM/PDMS films with different weight fractions. An LCR meter (E4980A, Agilent, Santa Clara, CA, USA) was employed for this purpose. A probe (16334A, Agilent, Santa Clara, CA, USA) with dimensions of 1.6 mm × 0.8 mm was placed in contact with the thin film, and measurements were conducted at frequencies of 200 Hz, 1 kHz, 10 kHz, 100 kHz, 1 MHz, and 2 MHz with a voltage level of 1 V.

#### 2.2.3. Solvent Evaporation Analysis 

Fourier transform infrared spectrometry (FT-IR) in attenuated total reflectance mode (ATR) was employed for analyzing the sample film. The measurement range was set at 4000 cm^−1^ to 600 cm^−1^ (FT/IR-4100 spectrometer, Jasco, Easton, MD, USA). Peaks associated with H_2_O and CO_2_ were eliminated from the analysis.

## 3. Results and Discussions

Figure 2a shows the surface of the Galinstan particle layer prior to the upper PDMS coating of each film fabricated for the TENG. The figure shows that the films with 2.22 wt.% and 5.17 wt.% Galinstan have very few LM particles compared to the amount of PDMS, resulting in a transparent film with dispersed LM particles that appear gray in color. The film with 12.97 wt.% Galinstan has gray areas spread throughout the entire film, and the film with 23.08 wt.% Galinstan has an even higher concentration of LM spread over the entire surface. The film with 40.30 wt.% Galinstan shows the LM distributed throughout the entire film, resulting in an opaque film that does not allow light to pass through. Magnified optical images of each film are presented below. At low concentrations, the LM is not evenly distributed, but at concentrations of 12.97 wt.% or higher, the LM completely overturns the entire film.

When attempting to ultrasonicate gallium-based LM particles in PDMS solution, the high viscosity of PDMS makes it challenging to pulverize the particles without subjecting them to prolonged ultrasonication with high power. However, this causes the temperature to rise due to the energy being applied, and the PDMS hardens before LM particles can be fully pulverized, making it difficult to achieve the desired result. To address this issue, n-hexane was added to the solution during ultrasonication to reduce the viscosity of the PDMS and curing agent, facilitating the pulverization of the LM particles [19]. Ultrasonication alone was not sufficient due to the high viscosity of PDMS, which prevented the ultrasonic waves from reaching the LM particles. Therefore, n-hexane, an organic solvent, was used to reduce the viscosity of the PDMS and curing agent. It was found that n-hexane was volatile and evaporated during the PDMS curing process, leaving no residue behind. As shown in Figure 2b, the FT-IR spectra of untreated PDMS and PDMS mixed with n-hexane were compared, and it was found that the peaks corresponding to Si-O-Si symmetric stretching (1074–1020 cm^−1^) [20] were at the same position in both films, while the peaks corresponding to n-hexane’s C-H stretching vibration (2940–2880 cm^−1^) and C-H deformation vibration (1480–1365 cm^−1^) were not visible, indicating that n-hexane was almost completely removed from the film.

To confirm the size distribution of Galinstan particles produced by ultrasonication, films were produced using various weight ratios, and a film with 23.08 wt.% was selected for analysis of the particle size distribution, as shown in Figure 2c. The inset of Figure 2c shows the optical image of this film. Since the particles are formed by crushing in a PDMS solution, conventional methods cannot be used to determine the particle size. Therefore, the size of the particles formed in the PDMS film was measured via optical microscopy, and representative values were set at intervals of 0.2 μm to count the number of particles within a certain area and plot a graph. As shown in the particle distribution in Figure 2c, the abundance of particles ranging from 0.4 to 1.2 μm was high, and most particles were smaller than 4 μm. The average value was measured to be 1.33 μm, and the median value was 1.12 μm, confirming that the particles were formed within a few micrometers through ultrasonication.

Figure 3a depicts a schematic diagram that elucidates the functioning of a triboelectric generator device, which employs a PDMS film containing Galinstan particles and a Cu film comprising an anode and a cathode. The TENG operates by combining contact electrification and electrostatic induction, resulting in the transfer of charge between the triboelectric layers [3]. The tribo-charging process generates opposite charges between the top electrode and the LM/PDMS layer, which reach their maximum values. Releasing the two surfaces generates a positive voltage peak. Compressing the copper electrode again results in the loss of the previously obtained stable state, generating voltage pulses in the opposite direction. This back-and-forth motion process is repeated, generating continuous voltage pulses.

In the conductor–dielectric mode with a vertical contact-separation method of the TENG, the metal layer serves as both the dielectric layer and electrode. As depicted in Figure 3b, a dielectric material is deposited on top of Metal B, and when it comes into contact with Metal A, polarization takes place, resulting in the formation of friction charges (−σ) inside the dielectric layer’s surface [21]. Upon separation, a potential difference arises between the two electrodes, and equal amounts of opposite charges migrate to each electrode. According to a previous report [22], the capacitance (C) of the tribo-dielectric layer in the triboelectric device can be expressed as follows:(1)$$C=\frac{\varepsilon_{o}S}{d_{0}+x\left(t\right)}$$
where *d*_0_ is the effective thickness of the dielectric layer, $\varepsilon_{0}$ is the vacuum permittivity, S is the area of the electrode, and *x*(*t*) is the distance between the two frictional dielectric layers upon separation. The effective dielectric thickness, *d*_0_, is defined as the sum of all the dielectric layer thicknesses, d_i_, divided by their respective relative permittivities, ε_r,i_.

The insertion of a layer with a higher relative permittivity, when the thickness of the composite film is fixed, causes a decrease in *d*_0_, as described by Equation (1). This decrease in *d*_0_ results in an increase in capacitance (C), as confirmed by the equation. Ultimately, this increase in capacitance can be attributed to the increase in the relative permittivity of the dielectric material, which results from the combination of PDMS and LM in the composite. It can be observed from this equation that if the relative permittivity is high, a high voltage and current can be generated using the vertical contact-separation method of the TENGs.

The capacitance of LM/PDMS films with varying weight percentages was determined by scanning in a frequency range from 200 Hz to 2 MHz, as shown in Figure 4a. The graph indicates that the capacitance of the film increased as the amount of LM particles increased, with the LM/PDMS film having a weight percentage of 23.08% exhibiting the highest capacitance value. These findings suggest that the LM/PDMS film with a weight percentage of 23.08% has the highest relative permittivity.

Figure 4b,c demonstrate the changes in the open-circuit voltage (Voc) and short-circuit current (Isc) for different weight percentages of LM/PDMS. The TENG produced with a pristine PDMS film had a Voc of 203.5V. The resulting TENGs had Voc values of 158.0 V, 160.0 V, 274.0 V, 337.5 V, and 171.8 V when the weight percentages of LM/PDMS were 2.22 wt.%, 5.17 wt.%, 12.97 wt.%, 23.08 wt.%, and 40.30 wt.%, respectively. Additionally, the Voc of the TENG produced with a pristine PDMS film was 203.5V, while the Isc values of the resulting TENGs were 13.9 μA, 16.3 μA, 23.8 μA, 26.2 μA, and 16.5 μA when the weight percentages of LM/PDMS were 2.22 wt.%, 5.17 wt.%, 12.97 wt.%, 23.08 wt.%, and 40.30 wt.%, respectively. Figure 4d shows the results of integrating the current graph by weight percentage for the charge amount. While the 2.22 wt.% and 5.17 wt.% devices demonstrated lower charge amounts than the pristine PDMS device, the other devices exhibited higher charge amounts. The 40.30 wt.% device showed a decrease in charge amount despite having a higher gallium weight compared to the 23.08 wt.% device. This result is consistent with the trend in the capacitance values of the films, as shown in Figure 4a, indicating that the TENG device produced with the LM/PDMS film having the highest relative permittivity showed the highest voltage, current, and charge amount.

In general, when friction occurs in a triboelectric component, charge accumulates across the entire surface due to opposite surface charges, which is called saturation. If air breakdown occurs beyond the saturation state, the surface free charge changes and moves to the oppositely charged particles in the air, achieving equilibrium, which is called charge recombination [23]. To prevent charge recombination, an intermediate transition layer can be inserted, which traps the surface free charge in the transition layer so that it does not move to the opposite component [24]. For this reason, Galinstan particles inserted in a PDMS film trap free charge and reduce charge recombination. As the amount of Galinstan within the PDMS film increases, the output voltage and current of the TENG also increase. The electrical characteristics of the TENG were found to be the highest in the component with a weight ratio of 23.08 wt.% of LM/PDMS and decreased in the component with a higher Galinstan weight ratio of 40.30 wt.% of LM/PDMS, according to the results of this experiment.

## 4. Conclusions

In conclusion, this study investigated the utilization of Galinstan particles in the production of TENGs for converting mechanical energy into electrical energy. The evaporation of the hexane solvent during the curing process resulted in a film containing variable concentrations of Galinstan particles. The addition of n-hexane during ultrasonication decreased the viscosity of the PDMS solution, facilitating the pulverization of LM particles. A film with a Galinstan concentration of 23.08 wt.% was chosen for analysis, and the particle size distribution was measured to be within a few micrometers through ultrasonication. The capacitance of the film increased as the quantity of LM particles increased, with the LM/PDMS film with a 23.08% weight percentage exhibiting the highest capacitance value. TENGs were constructed using LM/PDMS films with varying weight percentages and tested for open-circuit voltage, short-circuit current, and charge amount Q. The TENG with an LM/PDMS film with a 23.08% weight percentage had the highest relative permittivity and thus displayed the greatest voltage, current, and charge amount. This investigation confirms the correlation between the relative permittivity of the film and the performance of the component. Currently, there is active research in the practical applications of using TENGs with flexible and stretchable film properties for the production of wearable power sources and tactile sensors. Based on these findings, it is anticipated that TENGs produced using Galinstan particles in PDMS films will have potential applications in wearable devices, sensors, and biomedical fields. 

## Figures and Tables

**Figure 1 nanomaterials-13-01290-f001:**
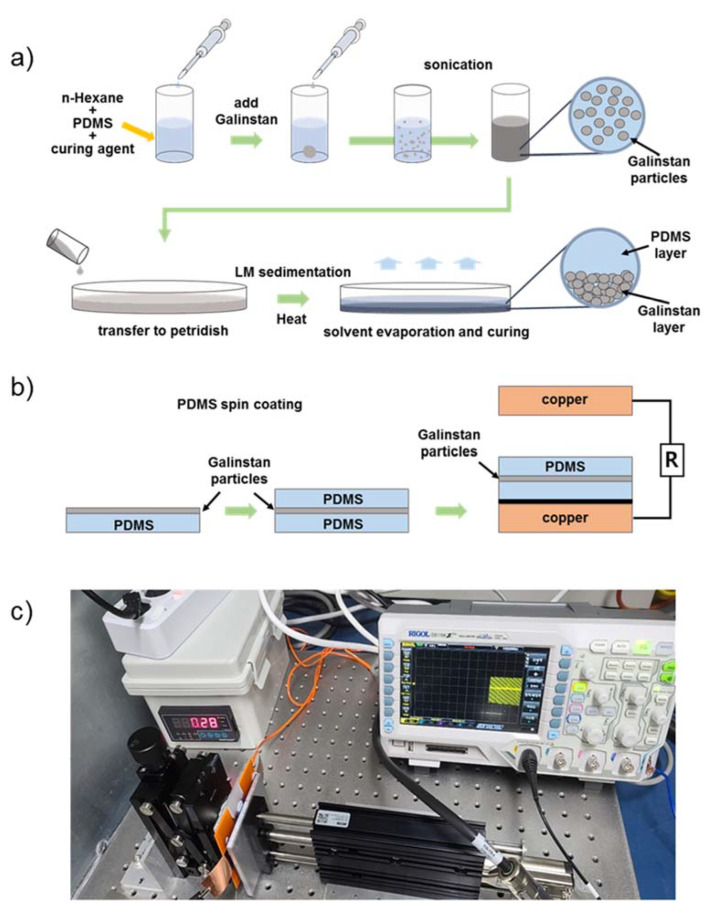
(**a**) Schematic illustration of the fabrication process of a TENG component using Galinstan particle layers. (**b**) Structure of a TENG with Galinstan particle layers inserted. (**c**) The experimental setup used to measure the TENG’s performance.

**Figure 2 nanomaterials-13-01290-f002:**
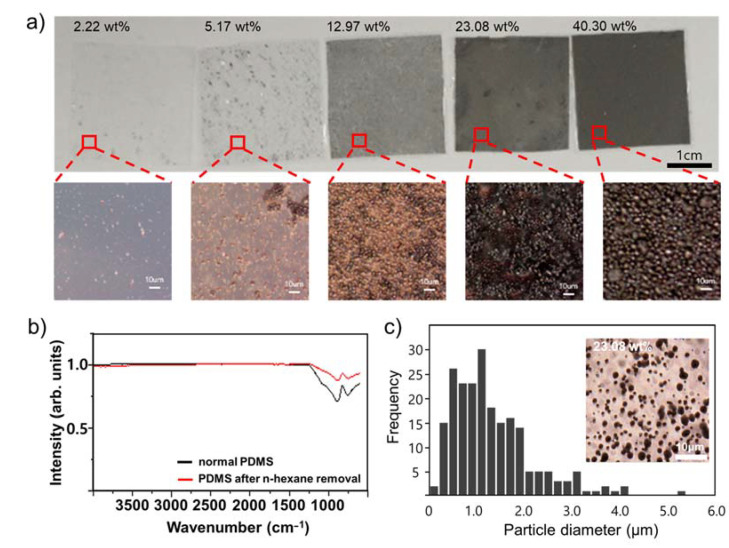
(**a**) Optical images of LM/PDMS dielectric film with weight percentages of 2.22 wt.%, 5.17 wt.%, 12.97 wt.%, 23.08 wt.%, and 40.30 wt.%, along with magnified images of each film. (**b**) FT-IR absorption spectra of 23.08 wt.% LM/PDMS film and normal PDMS film after thermal curing process. (**c**) Particle size frequency distribution curve analyzed through optical imaging of the 23.08 wt.% LM/PDMS film.

**Figure 3 nanomaterials-13-01290-f003:**
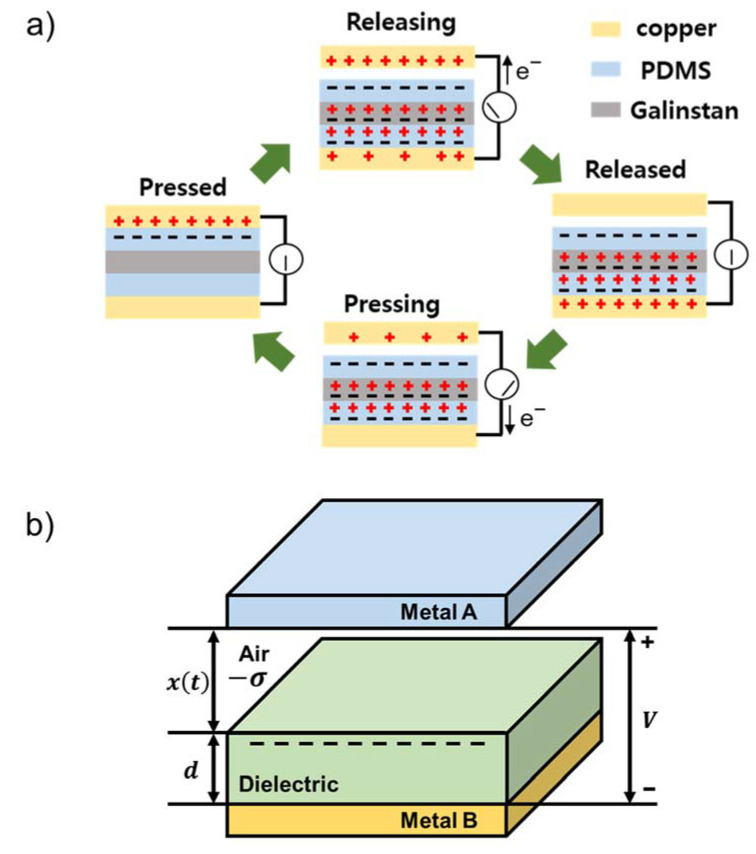
(**a**) The working principle of a TENG with LM/PDMS dielectrics. (**b**) Theoretical models for parallel-plate contact modes for a conductor–dielectric TENG.

**Figure 4 nanomaterials-13-01290-f004:**
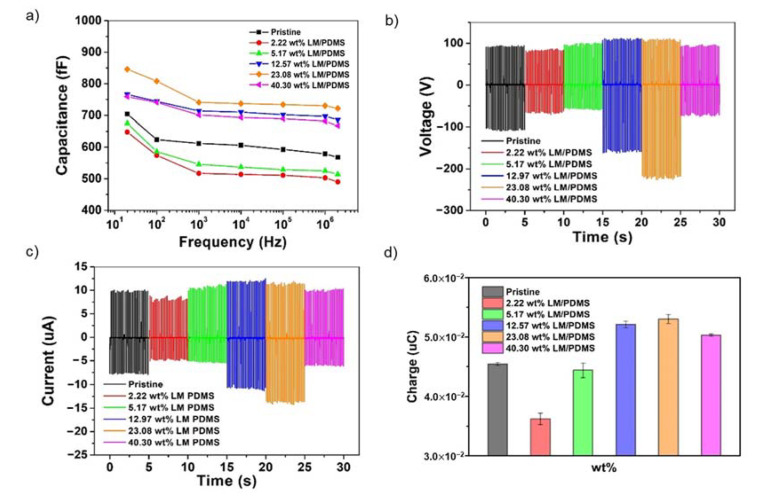
(**a**) Characteristic curves displaying capacitance variation as a function of frequency for different weight percentages. (**b**) Open-circuit voltage measurements obtained from TENGs constructed with LM/PDMS films of varying weight percentages. (**c**) Short-circuit current measurements obtained from TENGs constructed with LM/PDMS films of varying weight percentages. (**d**) The charge amount Q obtained from TENGs constructed with LM/PDMS films of varying weight percentages.

## Data Availability

Not applicable.

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
