# Peer review of "Amplifying the Output of a Triboelectric Nanogenerator Using an Intermediary Layer of Gallium-Based Liquid Metal Particles"

_nanomaterials, 2023, doi:10.3390/nano13071290_

Round 1
Reviewer 1 Report
The paper presents the results on fabrication and performance of a triboelectric nanogenerators using PDMS and liquid metal particles of galistan. The results are original and worth to be published. However, following recommendation should be considered and revision of the manuscript is required.
1. Filling polymers with metal or compound metal nanoparticles is a well-known strategy to increase a dielectric function of the medium. Indeed, using a liquid metal is an original approach but what are the advantages compared to the solid-state fillers? It is recommended to address this point in the Introduction, at least briefly. Also, to compare the performance of the NEGT presented in the current work to the performance (voltage, current, capacitance) obtained utilising solid-sate particles.
2. It would be nice to add a few sentences describing the optical measurements of the particle size into Section 2. Resolution of optical microscopy is limited by diffraction. Without special optical approaches it is hardly possible to find particle sizes below half of the wavelength. However, in Figure 2c, the particle sizes start from zero point. How is this possible?
3. Section describing principle of operation in lines 183-206 is trivial. It is recommended to shorten it and use some references to earlier reviews or first publications about TENG.
4. The part describing equations (1-3) is not original. It is more or less a shortened copy-past from ref. 22. These equations are difficult to understand without appropriate drawing. Subscripts “SC” and “OC” are not explained. These equations also do not play any role in the explanation of the obtained results. The whole section can be shortened to one sentence that an increase of relative permittivity by embedding the LM leads to the capacitance increase.
5. The discussions presented in lines 277-292 are the speculations, which are not supported by the experimental results. Since electron microscopy was not used, the authors disregarding a formation of nanoscale particles (not visible in optics). They could significantly contribute into the conductivity with increasing filling factor. They would also affect the percolation threshold. The authors do not address the formation of numerous interfaces between the LM and PDMS, which change the electronic structure of the composite. The suggestion about destroying the LM particles does not have any experimental proof. It is recommended to remove these physics-related, oversimplified and not experimentally or theoretically grounded speculations.
There is a couple of recommendations regarding the manuscript structure.
- Lines 116-120 can be incorporated into section 2.1;
- The explanation about compaction and change of wt% in lines 132-140 would better fit also to section 2.1.
A few comments about formulations:
- In line 40, it is not quite clear what the authors mean under “relatively polar and non-polar dielectric constants”;
- In line 56, “freezing” should be substituted by “freezing point of water” or 0 Celsius;
- In line 143-144, phrase “scattered gray LM particles” should be reformulated in a more scientific way;
- In line 175, it is recommended to substitute “frequency” by “abundance”;
- In lines 266-267, the statement “the surface free charges changes and moves to opposite component” is not clear. Do you mean an electric shortcut?
Reviewer 2 Report
The manuscript entitled “Amplifying the output of a triboelectric nanogenerator using an intermediary layer of gallium-based liquid metal particles”. The authors have fabricated the friction layer of the triboelectric nanogenerator by using gallium-based liquid metal/PDMS film. Material characterization of the LM/PDMS was studied such as optical image, FT-IR and particle size frequency distribution in this manuscript. In addition, the output electrical properties are studied. However, there are some points that are not clear. Therefore, I recommend the publication of this paper after major revision.
1. The author has reported preparing the dielectric film of LM/PDMS through layer by layer method for the TENG application. Why didn’t the author prepare the film just by the simple blending method?
2. Compared to the pristine PDMS film, the capacitance of 2.22 wt% and 5.17 wt% of LM/PDMS is lower. Why?
3. Would the sample has a leakage problem when applying the compressive force during the measurement as the LM is the liquid form?
4. It would be great if the authors could provide the practical application for this study.
